

# Cortinarius subsalor and C. tibeticisalor spp. nov., two new species from the section Delibuti from China

Meng-Le Xie[1,2,*], Jun-Liang Chen[3,*], Chayanard Phukhamsakda[2], Bálint Dima[4], Yong-Ping Fu[2], Rui-Qing Ji[2], Ke Wang[5], Tie-Zheng Wei[5] and Yu Li[1,2]

[1] Life Science College, Northeast Normal University, Changchun, Jilin, China
[2] Engineering Research Center of Edible and Medicinal Fungi, Ministry of Education, Jilin Agricultural University, Changchun, Jilin, China
[3] Science and Technology Research Center of Edible Fungi, Lishui, Zhejiang, China
[4] Department of Plant Anatomy, Institute of Biology, Eötvös Loránd University, Budapest, Hungary
[5] State Key Laboratory of Mycology, Institute of Microbiology, Chinese Academy of Sciences, Beijing, China
* These authors contributed equally to this work.

## ABSTRACT

*Cortinarius subsalor* and *C. tibeticisalor*, belonging to the section *Delibuti*, are described from China as new to science. *Cortinarius subsalor* has been found to be associated with *Lithocarpus* trees in subtropical China and resembling *C. salor*, but it differs from the later by having slender basidiomata and the narrower basidiospores. *Cortinarius tibeticisalor* was collected from eastern Tibetan Plateau, associated with *Abies*. It differs from other species within sect. *Delibuti* by having olive tinge of mature or dried basidiomata and bigger basidiospores. The molecular data also support *C. subsalor* and *C. tibeticisalor* as new species. The phylogenetic analyses and biogeography of sect. *Delibuti* are discussed and a key to the species of this section currently known in the world is provided.

## INTRODUCTION

*Cortinarius* (Pers.) Gray is an ectomycorrhizal fungal genus, associated with a wide host range of plants, such as Betulaceae, Caesalpiniaceae, Cistaceae, Dipterocarpaceae, Fagaceae, Myrtaceae, Pinaceae, Rhamnaceae, Rosaceae, Salicaceae and some herbaceous plants (*Frøslev, Brandrud & Jeppesen, 2006*; *Niskanen, 2008*). The genus is distributed worldwide with nearly 3,000 species (*Niskanen et al., 2018*; *Ammirati et al., 2021*; *Bidaud et al., 2021*). Even though it is the largest genus among macrofungi, its species diversity is still unclear. Most of *Cortinarius* species were originally discovered from Europe and America but rarely in Asia and Africa (*Horak, 1983*; *Garrido-Benavent et al., 2020*; *Xie et al., 2020*). Several systems of subgenus and sections in *Cortinarius* are erected based on the macromorphology of geographically limited samplings, but these are not supported by phylogenetic studies (*e.g. Fries, 1838*; *Trog, 1844*; *Orton, 1955*;

Corresponding authors
Tie-Zheng Wei, weitz@im.ac.cn
Yu Li, liyu@jlau.edu.cn

*Bidaud, Moënne-Loccoz & Reumaux, 1994*; *Garnica, Weiß & Oberwinkler, 2003*; *Garnica et al., 2005*; *Harrower et al., 2011*; *Stensrud et al., 2014*; *Niskanen et al., 2015*; *Garnica et al., 2016*; *Soop et al., 2019*). For example, *Garnica et al. (2005)* proposed natural classification system in *Cortinarius* involving the taxonomic rearrangement of the species into eight informal clades. *Soop et al. (2019)* presented a section-based taxonomy of *Cortinarius* based on four loci of a large global sampling.

*Cortinarius* sect. *Delibuti* (Fr.) Sacc. with characteristics of viscid pileus and stipe, have usually been considered as a section in subg. *Myxacium* (Fr.) Trog (*Trog, 1844*; *Earle, 1902*; *Orton, 1955*; *Brandrud et al., 1989*; *Consiglio, Antonini & Antonini, 2003*). *Delibuti* species can easily be distinguished by the anomaloid appearances, mild taste and subglobose basidiospores from other myxacioid species (*Orton, 1955*; *Soop, 2014*). Section *Delibuti* was also considered to belong to subg. *Phlegmacium* (Fr.) Trog (*Bidaud, Moënne-Loccoz & Reumaux, 1992*; *Bidaud, Moënne-Loccoz & Reumaux, 1994*). Recently, *Soop et al. (2019)* treated sect. *Delibuti* among anomaloid sections, not in myxacioid sections based on the shared characters of sect. *Delibuti* and sect. *Anomali* Konrad & Maubl., together with support in the phylogenetic analyses. In the past, numeral species were assigned to section *Delibuti* (*Fries, 1838*; *Earle, 1902*; *Bidaud, Moënne-Loccoz & Reumaux, 1992*; *Soop, 2013*; *Soop, 2014*); however, most species have been confirmed not to belong to this section (*Orton, 1955*; *Consiglio, 2012*; *Dima et al., 2016*; *Soop et al., 2019*). *Soop et al. (2019)* defined only ten species in sect. *Delibuti*, but the phylogenetic studies showed that the species diversity of this section is still unrevealed (*Harrower et al., 2011*; *Garnica et al., 2016*; *Soop et al., 2019*).

In China, over 237 *Cortinairus* species, including several new species, have been described from China (*Wei & Yao, 2013*; *Xie et al., 2019*; *Xie et al., 2020*; *Xie et al., 2021*; *Yuan et al., 2020*; *Luo & Bau, 2021*). Four species within sect. *Delibuti*, *C. betulinus* J. Favre from Heilongjiang, *C. delibutus* Fr. from Heilongjiang, Jilin, Qinghai, Sichuan and Yunnan, *C. illibatus* Fr. from Ningxia, and *C. salor* Fr. from Heilongjiang, Jilin, Liaoning and Inner Mongolia, were reported (*e.g.* *Teng, 1963*; *Yuan & Sun, 1995*; *Shao & Xiang, 1997*; *Li & Azbukina, 2011*; *Xie, 2018*; *Wang et al., 2020*), but the occurrence of species in China is controversial due to the lack of voucher specimens.

In this study, we have conducted taxonomic and phylogenetic studies of *Cortinairus* in China. Some glutinously violet *Cortinarius* specimens resembling *C. salor* were found during the intensive field work but during the identification process they turned out to be new species which we describe here based on morphological and ecological characteristics, as well as phylogenetic analyses evidences. We also discuss the phylogenetic relationship and biogeography of sect. *Delibuti*. A key is provided the species of sect. *Delibuti*.

## MATERIALS & METHODS

### Specimens and morphological description

Specimens were collected from Zhejiang Province and Tibet Autonomous Region, respectively. The collection sites in Zhejiang are the subtropical areas with the evergreen broadleaf forests dominated by *Lithocarpus brevicaudatus*. Meanwhile, the collection sites

in Tibet are the plateau-alpine areas with coniferous forests dominated by *Abies georgei* var. *smithii*. Fresh basidiomata were photographed in the field. Dried specimens were deposited in the Herbarium of Mycology, Jilin Agricultural University (HMJAU), Changchun, China. Macroscopic characteristics were measured and recorded for every basidiomata and color codes followed *Kornerup & Wanscher (1978)*. Microscopic features were examined and described in 5% KOH, Congo Red or Melzer's reagent and observed using a Zeiss AX10 light microscope. Thirty to forty mature basidiospores were measured (excluding apiculus and ornamentation) per collection. Q = variation in the L/W ratios between the specimens studied. $X_{av.}$ and $Q_{av.}$ = average value of basidiospores of per specimen.

## Phylogenetic reconstruction

DNA extraction, PCR amplifications, and sequencing methods followed *Xie et al. (2019)* and *Guan & Zhao (2020)*. The primers ITS1F and ITS4 were used amplification of nrDNA ITS region (*White et al., 1990*; *Gardes & Bruns, 1993*). The newly generated ITS sequences were submitted to GenBank. The ITS sequences for the phylogenetic analyses were selected based on results of BLASTn (>90% identity) in GenBank and UNITE and followed the publication by *Garnica et al. (2016)* and *Soop et al. (2019)*. Two species in section *Cyanites* Nespiak were chosen as outgroup followed *Xie et al. (2021)*.

Sequences (Table 1) for the phylogenetic analyses were aligned and edited with BioEdit 7.1.3.0 and Clustal X (*Thompson et al., 1997*; *Hall, 1999*). For phylogenetic analyses, Bayesian Inference (BI), Maximum Likelihood (ML) and Maximum Parsimony (MP) methods were implemented in this study. MrModeltest 2.3 was used to calculate the best model (HKY+I+G) for BI analysis (*Nylander et al., 2008*). The BI analysis was performed with MrBayes 3.2.6 (*Ronquist & Huelsenbeck, 2003*). Four Markov chains were run for 500,000 generations until the split deviation frequency value < 0.01, and sampled every 100th generation. The posterior probability values were estimated from the samples after discarding the first 25% (1,250) generations. A 50% majority rule consensus tree of all remaining trees were calculated. RAxML v. 1.5, implemented in raxmlGUI, were used to construct a ML tree, with a rapid bootstrapping algorithm involving 1,000 replicates (*Silvestro & Michalak, 2012*; *Stamatakis, 2014*). All parameters in the ML analysis were kept as defaults except for GTRGAMMA were chose as the model. The MP analysis was conducted in MEGA X (*Kumar et al., 2018*). The most parsimonious tree with length = 1,012 is shown. The consistency index is (0.442350), the retention index is (0.708912), and the composite index is 0.355860 (0.313587) for all sites and parsimony-informative sites (in parentheses). The bootstrap test was performed 1,000 replicates (*Felsenstein, 1985*). The MP tree was obtained using the Tree-Bisection-Regrafting (TBR) algorithm (*Nei & Kumar, 2000*) with search level 3 in which the initial trees were obtained by the random addition of sequences (10 replicates). The phylogenetic trees were visualized in FigTree 1.4.3. The Bayesian posterior probabilities values (BPP) ≥ 0.95, ML bootstrap values (ML) ≥ 75% or MP bootstrap values (MP) ≥ 75% are shown on the branches at the nodes (BPP/ML/MP).

**Table 1 ITS sequences used in the phylogenetic analyses.**

| Species | Voucher | Locality | Accession No. | References |
|---|---|---|---|---|
| *C. acutovelatus* | F16388 (UBC) | Canada | FJ039609 | *Harrower et al. (2011)* |
| *C. albocyaneus* Epitype | CFP1177 (S) | Sweden, Jämtland | KX302206 | *Dima et al. (2016)* |
| *C. alpinus* | HMJAU44407 | China, Inner Mongolia | MW911727 | This study |
| *C. anomalus* Neotype | CFP1154 (S) | Sweden, Ångermanland | KX302224 | *Dima et al. (2016)* |
| *C. basipurpureus* | PERTH 04259629 | Australia | AY669607 | *Garnica et al. (2005)* |
| *C. bolaris* | TUB 0118524 | Germany | AY669596 | *Garnica et al. (2005)* |
| *C. boreicyanites* Holotype | CFP931 (S) | Sweden, Jämtland | NR130214 | *Liimatainen et al. (2014)* |
| *C. calaisopus* | 60224 (OTA) | New Zealand | MN846380 | GenBank |
| *C. calaisopus* Holotype | PDD 94050 | New Zealand, Dunedin | NR157880 | GenBank |
| *C. camphoratus* | DAVFP26155 | Canada | EU821659 | *Harrower et al. (2011)* |
| *C. camphoratus* | SMI193 | Canada | FJ039626 | *Harrower et al. (2011)* |
| *C. carneoroseus* | EN76 (CORD) | Argentina | JX983157 | GenBank |
| *C. collinitus* | IB 19940257 | Sweden | AY033096 | *Peintner et al. (2002)* |
| *C. croceocoeruleus* | TUB 011833 | Germany | AY669590 | *Garnica et al. (2005)* |
| *C. cyanites* Neotype | AT2005069 (UPS) | Sweden, Uppland | NR130233 | *Liimatainen et al. (2014)* |
| *C. cypripedi* Holotype | PDD 107723 | New Zealand, Otago | KT875199 | *Soop (2016)* |
| *C. cystidiocatenatus* | HO A20518A6 | Australia, Tasmania | AY669651 | *Garnica et al. (2005)* |
| *C. delibutus* | F17048 (UBC) | Canada | FJ717515 | *Harrower et al. (2011)* |
| *C. delibutus* | SAT01-301-12 | USA | FJ717513 | *Harrower et al. (2011)* |
| *C. durifoliorum* Holotype | PDD 101829 | New Zealand, Westland | KJ635210 | *Soop, Wallace & Dima (2018)* |
| *C. eunomalus* | PDD 107706 | New Zealand | KT875201 | GenBank |
| *C. illibatus* | HMJAU48760 | China, Heilongjiang | MW911735 | This study |
| *C. illibatus* | AT2004220 (UPS) | Sweden | UDB002173 | UNITE |
| *C. illitus* Holotype | IB 19630414 | Argentina | AF389128 | *Peintner, Moncalvo & Vilgalys (2004)* |
| *C. illitus* | MQ19-CMMF003109 | Canada, Quebec | MN751331 | GenBank |
| *C. illuminus* Neotype | F44877 (S) | Sweden | KP866156 | *Niskanen et al. (2015)* |
| *C. khinganensis* Holotype | HMJAU44507 | China, Inner Mongolia | MT299952 | *Xie et al. (2021)* |
| *C. microglobisporus* Holotype | IB 20110123 | Italy | NR153027 | *Peintner et al. (2014)* |
| *C. obtusus* | SAT00-298-30 | USA | FJ717550 | *Harrower et al. (2011)* |
| *C. phlegmophorus* | Typus-M3 | India | AY083186 | *Peintner et al. (2003)* |
| *C. pluvius* | HMJAU44391 | China, Inner Mongolia | MW911726 | This study |
| *C. porphyroideus* | 61406 (OTA) | New Zealand | JX178612 | *Teasdale et al. (2013)* |
| *C. pseudocandelaris* | F17165 OC93 (UBC) | Canada, BC | GQ159908 | *Harrower et al. (2011)* |
| *C. psilomorphus* Holotype | PDD 103885 | New Zealand | KF727393 | *Soop (2016)* |
| *C. putorius* Holotype | TN 07-411 (H) | USA | NR153038 | *Ariyawansa et al. (2015)* |
| *C. pyrenaicus* | JB-8573/15 | Spain, Gisclareny | KX239900 | *Cadiñanos, Gomez & Ballarà (2016)* |
| *C. rattinoides* Holotype | PDD 88283 | New Zealand | JX000375 | GenBank |
| *C. rotundisporus* | PERTH 05255074 | Australia | AY669612 | *Garnica et al. (2005)* |
| *C. rotundisporus* | G12 | Australia | AF136738 | *Sawyer, Chambers & Cairney (1999)* |
| *C. salor* | IB 19940297 | Austria | UDB001066 | *Peintner et al. (2001)* |
| *C. salor* | TUB 011838 | Germany | AY669592 | *Garnica et al. (2005)* |
| *C. salor* II | TUF106868 | Estonia | UDB011268 | UNITE |
| *C. salor* II | TAAM128516 | Estonia | UDB015945 | UNITE |

| Species | Voucher | Locality | Accession No. | References |
|---|---|---|---|---|
| *C. septentrionalis* | ARAN Fungi03516 | Sweden, Harjedalen | KX239915 | *Cadiñanos, Gomez & Ballarà (2016)* |
| *C. spilomeus* Neotype | TEB CFP1137 (S) | Sweden | KX302267 | *Dima et al. (2016)* |
| *C. stillatitus* | TUB 011587 | Germany | AY669589 | *Garnica et al. (2005)* |
| **C. subsalor** | **HMJAU48758** | **China, Zhejiang** | **MW911733** | **This study** |
| **C. subsalor** Holotype | **HMJAU48759** | **China, Zhejiang** | **MW911734** | **This study** |
| **C. subsalor** | **MHHNU 30409** | **China, Hunan** | **MK250915** | **GenBank** |
| *C. suecicolor* Holotype | PDD 74698 | New Zealand | JX000360 | GenBank |
| *C. tabularis* Epitype | CFP949 (S) | Sweden | KX302275 | *Dima et al. (2016)* |
| *C. tasmacamphoratus* | HO A20606A0 | Tasmania | AY669633 | *Garnica et al. (2005)* |
| *C. tessiae* | PDD 94054 | New Zealand, Dunedin | JQ287698 | GenBank |
| *C. tessiae* | PDD 72611 | New Zealand | HM060317 | GenBank |
| **C. tibeticisalor** | **HMJAU48761** | **China, Tibet** | **MW911731** | **This study** |
| **C. tibeticisalor** | **HMJAU48762** | **China, Tibet** | **MW911732** | **This study** |
| **C. tibeticisalor** | **HMJAU48763** | **China, Tibet** | **MW911730** | **This study** |
| **C. tibeticisalor** Holotype | **HMJAU48764** | **China, Tibet** | **MW911729** | **This study** |
| *C. vanduzerensis* | VMS28 | Canada | FJ717562 | *Harrower et al. (2011)* |
| *C. vibratilis* | IB 19970078 | USA | AF325584 | *Peintner et al. (2001)* |
| *C.* sp. | CSU CO 2476 | Colombia, Antioquia | MF599228 | GenBank |
| *C.* sp. | FLAS-F-60161 | USA | MF153022 | GenBank |
| *C.* sp. | YM714 | Japan, Hokkaido | LC175538 | GenBank |
| *C.* sp. | 1780 | Italy | JF907917 | *Osmundson et al. (2013)* |
| *C.* sp. | SWUBC500 | Canada | DQ481723 | *Wright, Berch & Berbee (2009)* |
| *C.* sp. | PDD 72685 | New Zealand | MH101524 | GenBank |

**Note:**
New species is in bold.

## Nomenclature

The electronic version of this article in Portable Document Format (PDF) will represent a published work according to the International Code of Nomenclature for algae, fungi, and plants, and hence the new names contained in the electronic version are effectively published under that Code from the electronic edition alone. In addition, new names contained in this work have been submitted to MycoBank from where they will be made available to the Global Names Index. The unique MycoBank number can be resolved and the associated information viewed through any standard web browser by appending the MycoBank number contained in this publication to the prefix "http://www.mycobank.org/MycoTaxo.aspx?Link=T&Rec=". The online version of this work is archived and available from the following digital repositories: PeerJ, PubMed Central, and CLOCKSS.

## RESULTS

### BLASTn results

The BLASTn against GenBank and UNITE databases taked the holotype specimens as the examples. The BLASTn results showed that these two new species distinct from other
members of *Cortinarius* and close to sect. *Delibuti*. The ITS sequence of *C. subsalor* (MW911734, holotype) has 99% identity with *C. salor* (MK250915). Here we addressed it as *C. subsalor*. The percent identity of *C. subsalor* with *C. salor* s.l. (AY669592, UDB015945) and *C. delibutus* (FJ717515) are 96% and 95%, respectively. The ITS sequence of *C. tibeticisalor* (MW911729, holotype) has 93%, 91%, 90% identity with *Cortinarius* sp. (LC098750), *C. delibutus* (FJ717515) and *C. tessiae* (JQ287698), respectively.

## Phylogenetic analyses

The matrix contained 66 ITS sequences with 767 nucleotide sites is available from TreeBASE under S28399 (https://www.treebase.org/treebase-web/search/study/summary.html?id=28399). The BI, ML and MP results showed similar topologies and ML tree was selected as the backbone phylogeny (Fig. 1). The phylogenetic analyses showed 11 sections including one singleton species from Argentina, and two singletons from New Zealand. Every section formed separate monophyletic lineages with strong statistical support. Section *Delibuti* formed a distinct clade (BPP = 0.96) separate from other sections. Section *Delibuti* split into five main clades based on the analyses of ITS sequences. Clade A and B consist of Australasian species. Clade C is a clade including our new species from the Tibetan Plateau. Clade D consist of the species distributed in Europe, Asia and North and South America. Clade E represents species in the Northern Hemisphere. *Cortinarius subsalor* (BPP/ML/MP = 1.00/100%/100%, clade E) and *C. tibeticisalor* (BPP/ML/MP = 1.00/100%/100%, clade C) formed a dinstinct lineages with high statistical support, respectively. Furthermore, *C. subsalor* formed a sister relationship with the European *C. salor* (BPP = 0.97, clade E).

## Taxonomy

***Cortinarius subsalor* M.L. Xie, T.Z. Wei & Y. Li, sp. nov.**
MycoBank No. MB839320
(Fig. 2)

**Etymology.** The name refers to its affinity to *Cortinarius salor*.

**Holotype.** CHINA. Zhejiang: Baishanzu Mountain, Qingyuan county, on moist soil under *Lithocarpus brevicaudatus* (Fagaceae) forest with scattered Theaceae and *Rhododendron*, 27°45′44″N, 119°11′50″E, ASL 1,510 m, 20 July 2020, Jun-Liang Chen, *QY-0235(1192-1198)* (HMJAU48759), GenBank: MW911734.

**Diagnosis.** Pileus hemispherical to plane, violet, glutinous; lamellae violet at first, then turning pale grayish violet; stipe slender, pale violet, then brown; glutinous veil violet. Basidiospores on average 8.0–8.3 × 6.9–7.0 μm, subglobose to broadly ellipsoid. Differing from other species in sect. *Delituti* by the violet color of basidiomata, the distribution of subtropical China and association with *Lithocarpus brevicaudatus*.
**Description**

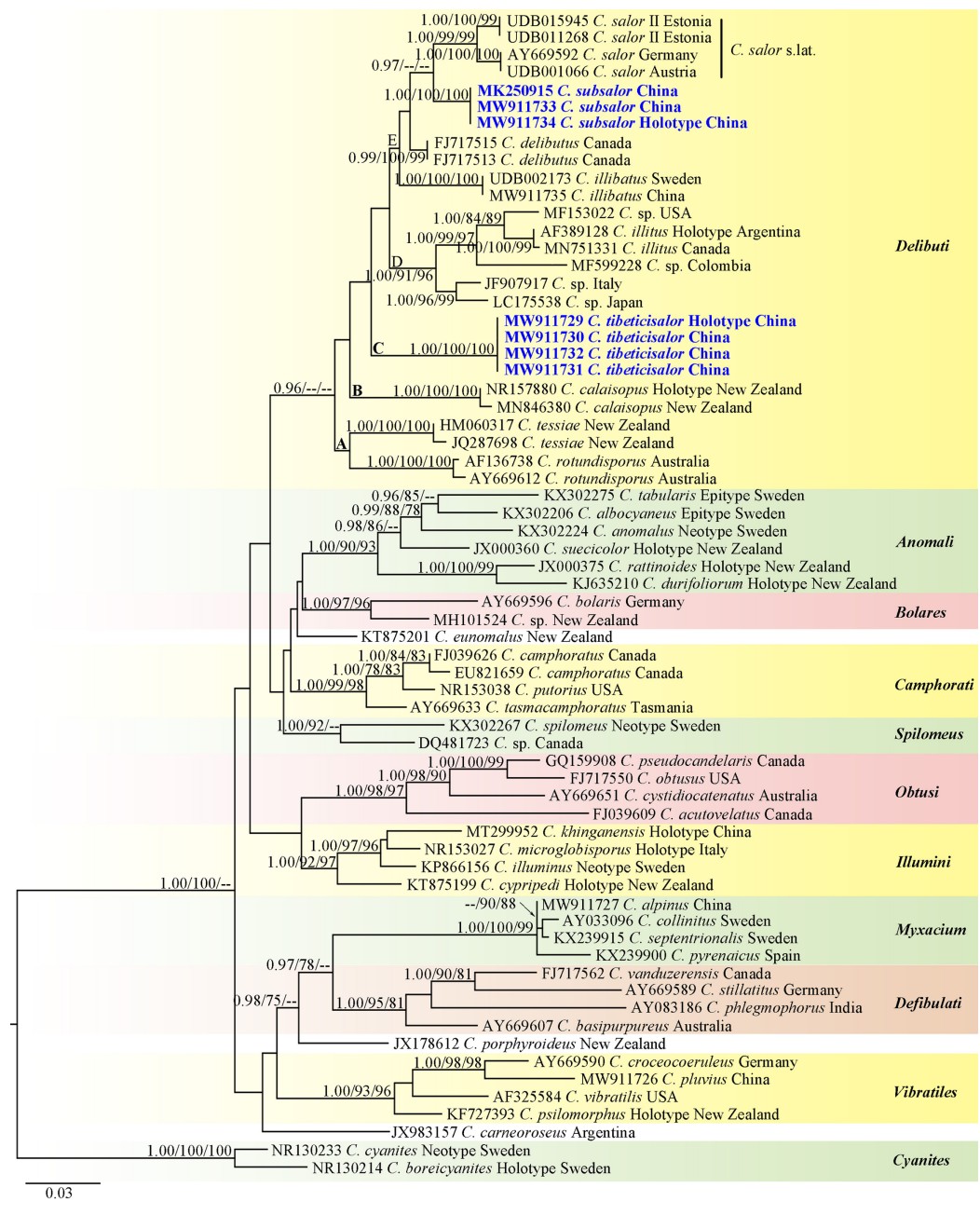

**Figure 1 ML phylogram inferred from nrDNA ITS sequence data.** The tree is rooted with sect. *Cyanites*. The Bayesian posterior probabilities (BPP) ≥ 0.95, ML bootstrap values (ML) ≥ 75% and MP bootstrap values (MP) ≥ 75% are shown on the branches (BPP/ML/MP). New species is marked by blue bold.

Pileus 20–50 mm, hemispherical at first, then convex to applanate; bluish violet (18B6-18C7) at first, purple (15B6-15C7) to purplish red (14A6-14B7) at the centre, then grayish violet (17B4-17C5), pale violet (19A3) at the margin; surface glutinous. Lamellae emarginate; moderately crowded; violet (17B6) when young, then grayish violet (17B4-17C5) to pale grayish violet (15B1-15C2); edge almost even. Stipe slender,

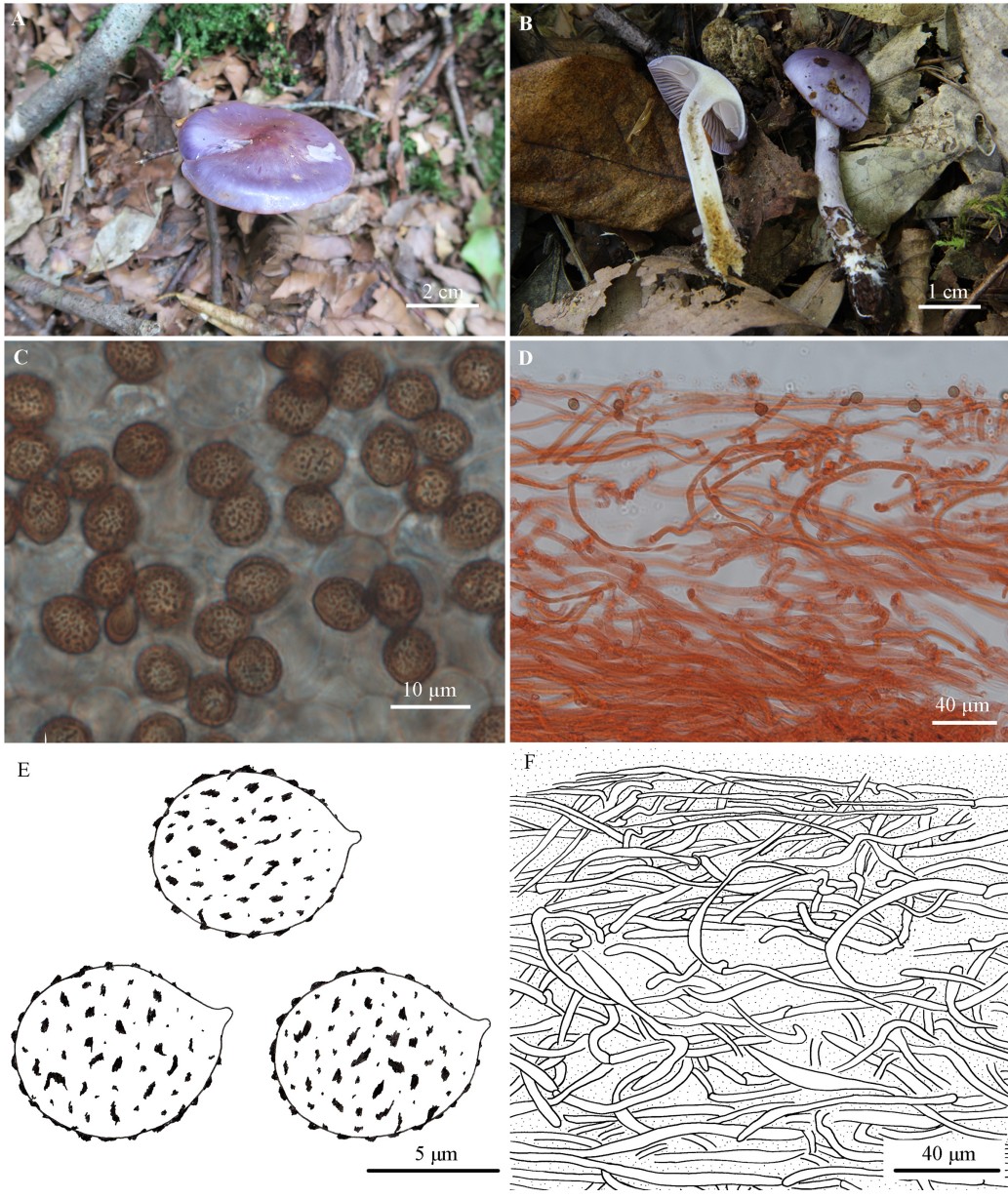

**Figure 2 *Cortinarius susalor.*** (A) HMJAU48759 (Holotype). (B) HMJAU48758. (C, E) Basidiospores (HMJAU48759). (D, F) Pileipellis (HMJAU48759). (Photo credits: (A) Jun-Liang Chen; (B)–(F) Meng-Le Xie).

35–65 mm long, 3–7 mm thick, clavate at base (up to 14 mm); pale violet to grayish violet (19A3-19B5), later whitish, lightly brown to brown (7D6-7E7); surface with viscid universal veil, basal mycelium white with bluish tinge. Universal veil viscid, violet, remnants forming a girdle on the upper part of the stipe, disappearing with age. Context whitish at the pileus, slightly with yellowish tinge at the center, pale violet tinge extend outward, hygrophanous near lamellae; white with pale violet tinge at the apex of the stipe, yellow at the lower part; somewhat hollow within stipe. Odor not significant, taste mild.

Basidiospores 7.7–9.5 (10.6) × 6.2–7.7 (8.7) µm, Q = 1.10–1.29 (holotype), $X_{av.}$ = 8.0–8.3 × 6.9–7.0 µm, $Q_{av.}$ = 1.20, subglobose to broadly ellipsoid, moderately coarsely verrucose, moderately dextrinoid. Basidia 4-spored. Lamellar edges fertile. Pileipellis: epicutis strongly gelatinous, about 180–250 µm thick, with hyphae 2–7 µm wide, yellowish to colorless in 5% KOH, some hyphae with small encrusted granules. Hypodermium present; hypodermial hyphae 4–10 µm wide, cylindrical, almost colorless in 5% KOH, smooth. Clamp connections present.

**Exsiccatae.** Pileus grayish violet (19B3-19C4) at the margin, light brown to dark brown (6D6-6F8) at the centre; lamellae rust brown (6E8); stipe brown (6D7-6E7), lighter downwards, yellowish white (4A2) at base.

**ITS sequence**. The ITS sequence of the holotype is distinct from other members of sect. *Delibuti* and deviating from them by at least 22 substitutions and indel positions.

**Ecology and distribution.** In subtropical evergreen broadleaf forests, associated with *Lithocarpus brevicaudatus* (Fagaceae). Known from Zhejiang and Hunan province of China.

**Additional specimens examined.** CHINA. Zhejiang: Baishanzu Mountain, Qingyuan county, on moist soil under *Lithocarpus brevicaudatus* (Fagaceae) forest with scattered Theaceae and *Rhododendron*, 27°45′55″N, 119°11′0″E, ASL 1500 m, 20 August 2020, Meng-Le Xie, *20xml12101* (HMJAU48758), GenBank: MW911733.

*Cortinarius tibeticisalor* **M.L. Xie, T.Z. Wei & Y. Li, sp. nov.**
MycoBank No. MB839321
(Fig. 3)

**Etymology.** The name refers to the Tibetan Plateau, the type locality, and its similarity to *C. salor*.

**Holotype.** CHINA. Tibet Autonomous Region: Sejila Mountain, Linzhi city, on moist soil in *Abies* forest with scattered *Rhododendron*, 29°35′26″N, 94°35′53″E, ASL 4120 m, 5 September 2020, Meng-Le Xie, *20xml12416* (HMJAU48764), GenBank: MW911729.

**Diagnosis.** Pileus hemispherical to applanate, violet, glutinous, margin wavy, somewhat olive when mature; lamellae for a long time violet, then pale grayish violet to violet gray; stipe robust, bluish gray to brown with olive tinge; veil glutinous, violet. Basidiospores on average 10.3–10.8 × 8.7–8.9 µm, subglobose to broadly ellipsoid, rarely ellipsoid. Differing from other species in sect. *Delituti* by the olive tinge of basidiomata and the large basidiospores.

**Description**
Pileus 50–85 mm, hemispherical at first, then convex to plane, sometimes slightly depressed, wavy at margin of mature basidiomata, violet (17C7) at first, especially at the centre, paler violet towards the margin, then grayish orange (5B5) to brown (5D6-5E7) with olive tinge, dark at the centre; surface glutinous. Lamellae emarginate, moderately

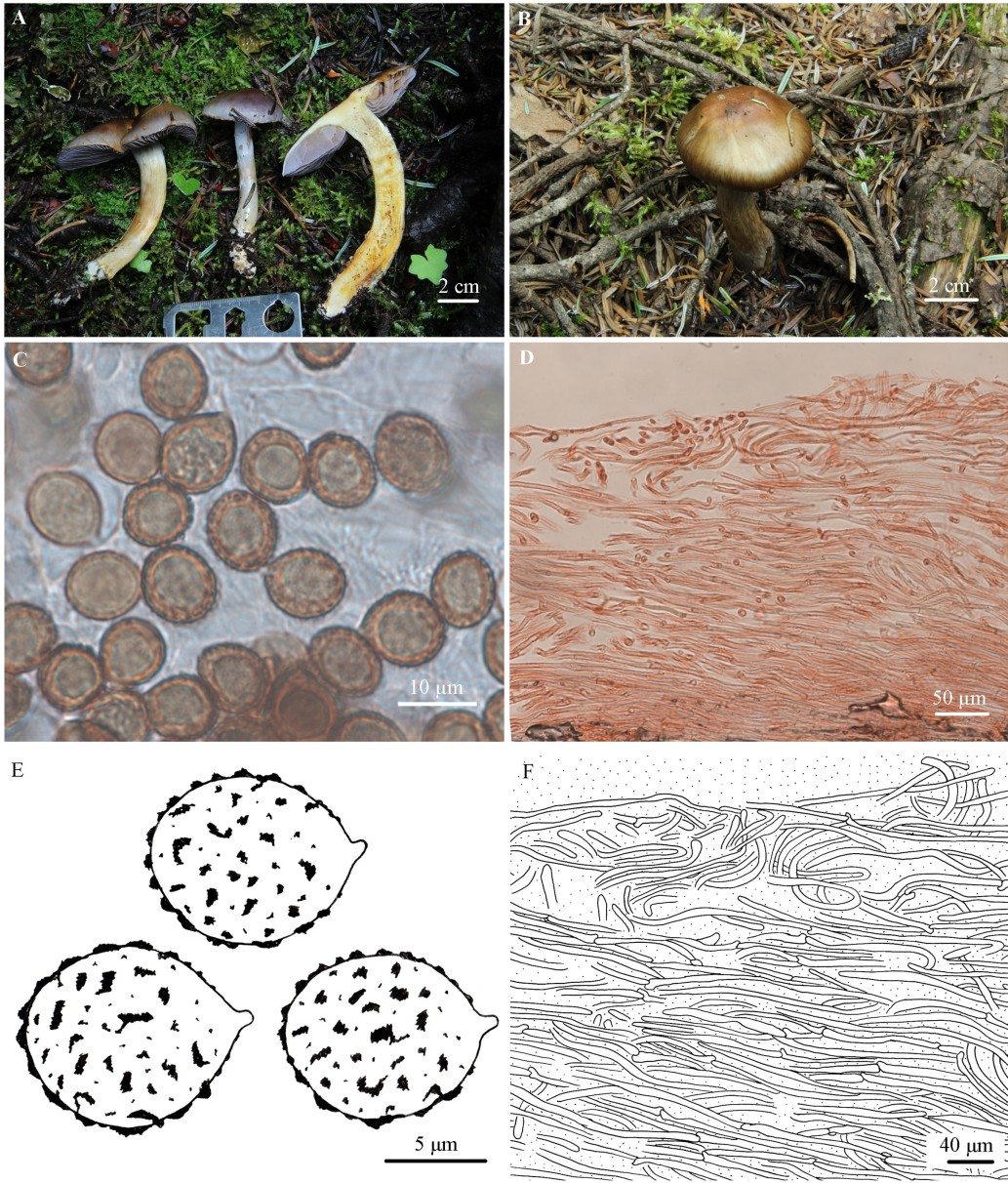

**Figure 3 *Cortinarius tibeticisalor*.** (A) HMJAU48764 (Holotype). (B) HMJAU48762. (C, E) Basidiospores (HMJAU48764). (D, F) Pileipellis (HMJAU48764). (Photo credits: Meng-Le Xie).

crowded, persistently violet (17C7), then grayish violet (19B4-19C6) to violet gray (19B2); edge uneven, slightly serrate. Stipe 85–120 mm long, 10–15 mm thick, clavate at base (up to 23 mm); surface with viscid bluish gray (19B2) universal veil remnants, then becoming yellow to brown with olive tinge (4B6-4D7), grayish violet (19B4-19C6) at the apex; basal mycelium white. Universal veil viscid, violet, remnants forming a girdle on the upper part of the stipe, dispearing with age. Context white with marbled violet tinge at first, slightly yellowish from the center of the pileus, then yellow at the stipe, especially at the middle. Odor weak when fresh, somewhat like honey when old or dry. Taste mild.

Basidiospores 9.7–10.9 (12.6) × 7.7–9.0 (10.0) μm, Q = 1.13–1.30 (holotype), $X_{av.}$ = 10.3–10.8 × 8.7–8.9 μm, $Q_{av.}$ = 1.20–1.23, subglobose to broadly ellipsoid, rarely ellipsoid, moderately coarsely verrucose, weakly dextrinoid. Basidia 4-spored. Lamellar edges fertile, with narrow clavate cells. Pileipellis: epicutis strongly gelatinous, about 300–410 μm thick, hyphae 3–8 μm wide, with yellowish intracellular pigment in 5% KOH, smooth. Hypodermium present, hyphae 7–15 μm wide, irregular, almost colorless in 5% KOH, smooth. Clamp connections present.

**Exsiccatae.** Pileus olive brown (4E6-4F7) at margin, yellowish brown (5D7-5E8) at centre; lamellae dark bluish gray (19E2-19F2); stipe bluish white at apex, light brown (6D6-6E7) to dark brown (6F4-6F8), white at base.

**ITS sequence.** The ITS sequence of the holotype is distinct from other members of sect. *Delibuti* and deviating from them by at least 40 substitutions and indel positions.

**Ecology and distribution.** In plateau-alpine coniferous forests, associated with *Abies* (Pinaceae) trees. Known from Tibetan Plateau of China.

**Additional specimens examined.** CHINA. Tibet Autonomous Region: Sejila Mountain, Linzhi city, on moist soil under *Abies* forest with scattered *Rhododendron*, 29°35′25″N, 94°35′55″E, ASL 4170 m, 28 August 2019, Meng-Le Xie, *19xml10976* (HMJAU48761), GenBank MW911731, *19xml10981* (HMJAU48762), GenBank MW911732; Sejila Mountain, Linzhi city, on moist soil under *Abies* forest with scattered *Rhododendron*, 29°35′26″ N, 94°35′53″E, ASL 4120 m, 5 September 2020, Meng-Le Xie, *20xml12395* (HMJAU48763), GenBank: MW911730.

## Key to species of sect. *Delibuti*

1 Distributed in Northern Hemisphere ............................................................... 2

- Distributed in Southern Hemiphere ................................................................ 9

2. Pileus usually yellowish to ochraceous without blue ................................... 3

- Pileus more or less violet to blue when young, sometimes partly yellow .............. 4

3. Lamellae usually blue when young, veil yellowish........................ ***C. delibutus***

- Lamellae pinkish ochraceous clay, veil not yellowish ...................... ***C. illibatus***

4. Pileus frankly blue when young, stipe bluish, veil violet ............................. 5

- Pileus grayish blue to olive brown, stipe pale, veil different .......................... 7

5. Basidiomata usually small, lamellae violet, then grayish to brownish, stipe usually slender (< 10 mm), base white with bluish tinge, basidiospores on average 8.0–8.3 × 6.9–7.0 μm, subglobose to broadly ellipsoid, distributed in subtropical China, associated with *Lithocarpus brevicaudatus* ................................................................. ***C. subsalor***

- Basidiomata usually bigger, lamellae persistently lilaceous or bluish, stipe usually more robust (> 10 mm thick)........................................................................ 6

6. Pileus usually staining buff or fading from the centre, stipe base usually grayish brown, basidiospores 7–9 × 6–8 µm, globose to subglobose, distributed in Europue, associated with deciduous and coniferous trees........................................... ***C. salor***

- Pileus usually olive brown when mature, stipe base usually white, basidiospores 10.3–10.8 × 8.7–8.9 µm, subglobose to broadly ellipsoid, rarely ellipsoid, distributed in Tibetan Plateau of China, associated with *Abies*................................***C. tibeticisalor***

7. Basidiomata small, pileus yellow to olive-ochre at the centre, grayish blue towards the margin, soon fading, veil yellow, basidiospores 7.5–9.5 × 6.5–7.5 µm, subglobose, associated with *Betula* ................................................. ***C. betulinus***

- Basidiomata robust, associated with coniferous forests ............................... 8

8. Pileus usually olive brown with a violet margin, veil olive brown, basidiospores 8–10 × 7–8 µm, globose, associated with *Picea* .....................................***C. transiens***

- Pileus not olive brown, but prefer orange tinge, basidiospores 7.5–9.5×6.5–7.5 µm, subglobose, usually associated with *Abies*, rarely occur in *Picea* forests  ***C. largodelibutus***

9. Associated with *Nothofagus* ...................................................... 10

- Associated with Myrtaceae trees ................................................... 11

10. Pileus viscid, blue-green to aerugineous, stipe blue green, basidiospores 6.5–8.5 × 6–7 µm, subglobose, destributed in Australasia .................................... ***C. tessiae***

- Pileus glutinous, greyish yellow to greyish orange, stipe violet, then becoming white to pale brownish, basidiospores ellipsoid, destributed in North and South America.......................................................................... ***C. illitus***

11. Basidiomata distinctly viscid to glutinous, mainly greyish blue-green, basidiospores 7–9 × 7–8 µm, globose to subglobose ..................................... ***C. rotundisporus***

- Basidiomata weakly viscid, stipe often dry, mainly yellow-green to olive, Veil orange to ochraceous, basidiospores 6–7.5 × 5.5–6.5 µm, subglobose................. ***C. calaisopus***

## DISCUSSION

*Cortinarius subsalor* is similar to *C. betulinus*, *C. salor* and *C. transiens* (Melot) Soop due to the bluish tinge of the basiodiomata. However, *C. betulinus* is usually grayish blue at the margin of the pileus and soon fading, the stipe is often pale and the veil usually is yellow (*Kibby, 2005*; *Niskanen et al., 2008*; *Soop, 2014*). The pileus of *C. transiens* has a violet tone towards the margin, while the centre is more olive gray to yellowish brown even in young specimens, the stipe is pale, and the gelatinous veil is olive brown (*Soop, 1990*, *2014*). In China, sometimes some bluish myxacioid species have been misidentified as *C. salor* (MHHNU30409, GenBank: MK250915), collected from Hunan Province. Our phylogenetic analyses showed that this sequence belong to the new species *C. subsalor*. *Cortinarius salor* has persistently lilaceous lamellae, the stipe is more robust (>10 mm thick) and the base is more grayish brown, the basidiospores are rounder (7–9 × 6–8 µm), and it occurs in European woodlands (*Orton, 1955*; *Consiglio, Antonini & Antonini, 2003*; *Soop, 2014*). Based on these features, *C. salor* can be distinguished from the Asian *C. subsalor*.

*Cortinarius tibeticisalor* is characterized by the basidiomata usually violet when young, then grayish orange to brown with an olive tinge, larger basidiospores and a restricted distribution in the Tibetan Plateau. *Cortinarius tibeticisalor* is similar to *C. salor* in young stage, however, the basidiospores ($7–9 \times 6–8$ μm) of *C. salor* are significantly smaller and rounder, and the basidiomata never have olive tinge (*Orton, 1955*; *Consiglio, Antonini & Antonini, 2003*; *Soop, 2014*).

According to our phylogenetic analyses, sect. *Delibuti* demonstrates a widely distributed lineage of *Cortinarius*, in both the Northern and Southern Hemispheres. This bihemispherical distribution is also seen in several other lineages in *Cortinarius*, such as *Anomali*, *Bolares*, *Camphorati*, *Defibulati*, *Illumini*, and *Vibratiles*, this is concordant with other studies (*e.g. Harrower et al., 2015*; *Garnica et al., 2016*; *Soop et al., 2019*). The nrDNA ITS region is not suitable to draw conclusions for comprehensive phylogenetic evaluation, however, there are some interesting patterns indicated in sect. *Delibuti* to be further discussed. The basal lineages (clade A and B) of *Delibuti* are solely distributed in the Australasia showing a presumable origin of this section in Australasia. Interestingly, clade D contains species from multiple continents in the Northern and Southern Hemispheres. Some species are distributed in Asia (*Cortinarius* sp., LC175538), in Europe (*Cortinarius* sp., JF907917), and South America, like *Cortinarius* sp. (MF599228) from Colombia and *C. illitus Moser & Horak (1975)* originally described from Argentina, but also found in North America (according to the sequences in GenBank).These patterns could explain that the evolution of sect. *Delibuti* is limited to the ectomycorrhizal host specificity, as well as geographic barriers (*Wang & Qiu, 2006*; *Brandrud, 1996*; *Wilson, Hosaka & Mueller, 2017*; *Feng et al., 2016*). The evolution and origin of sect. *Delibuti*, including the genus *Cortinarius* will be a subject for future research.

## ACKNOWLEDGEMENTS

The authors are grateful to the two reviewers, Tai-Hui Li and Chang-Lin Zhao, for their constructive comments and suggestions on the manuscript. We thank to Dr. Rong Xie and Ms. Hui-Juan Sun (Tibet Academy of Agricultural and Animal Husbandry Sciences) and Mr. Ju-Zuo Li (Life Science College, Northeast Normal University) for their kind help in the fieldwork. We also thank to Mr. Ji-Peng Li and Yang Wang (Engineering Research Center of Edible and Medicinal Fungi, Ministry of Education, Jilin Agricultural University) for their kind help in the molecular studies.

### Funding

This study was supported by the China Agriculture Research System (No. CARS20), Overseas Expertise Introduction Project for Discipline Innovation (No. D17014), National Natural Science Foundation of China (No. 31270072), the Special Funds for the Young Scholars of Taxonomy of the Chinese Academy of Sciences (No. ZSBR-001), the National Key Research and Development program of China (No. 2019YFC1604703). The work of Bálint Dima was supported by the ELTE Thematic Excellence Program 2020 (TKP2020-

IKA-05) financed by the National Research, Development and Innovation Office. There was no additional external funding received for this study. The funders had no role in study design, data collection and analysis, decision to publish, or preparation of the manuscript.

## Grant Disclosures
The following grant information was disclosed by the authors:
China Agriculture Research System: CARS20.
Overseas Expertise Introduction Project for Discipline Innovation: D17014.
National Natural Science Foundation of China: 31270072.
Special Funds for the Young Scholars of Taxonomy of the Chinese Academy of Sciences: ZSBR-001.
National Key Research and Development Program of China: 2019YFC1604703.
ELTE Thematic Excellence Program 2020: TKP2020-IKA-05.
National Research, Development and Innovation Office.

## Competing Interests
The authors declare that they have no competing interests.

## Author Contributions

- Meng-Le Xie conceived and designed the experiments, performed the experiments, analyzed the data, prepared figures and/or tables, authored or reviewed drafts of the paper, and approved the final draft.
- Jun-Liang Chen conceived and designed the experiments, performed the experiments, analyzed the data, prepared figures and/or tables, authored or reviewed drafts of the paper, and approved the final draft.
- Chayanard Phukhamsakda analyzed the data, prepared figures and/or tables, authored or reviewed drafts of the paper, and approved the final draft.
- Bálint Dima analyzed the data, prepared figures and/or tables, authored or reviewed drafts of the paper, and approved the final draft.
- Yong-Ping Fu analyzed the data, authored or reviewed drafts of the paper, and approved the final draft.
- Rui-Qing Ji conceived and designed the experiments, performed the experiments, authored or reviewed drafts of the paper, and approved the final draft.
- Ke Wang performed the experiments, analyzed the data, prepared figures and/or tables, and approved the final draft.
- Tie-Zheng Wei conceived and designed the experiments, performed the experiments, analyzed the data, authored or reviewed drafts of the paper, and approved the final draft.
- Yu Li conceived and designed the experiments, analyzed the data, authored or reviewed drafts of the paper, and approved the final draft.

## DNA Deposition

The following information was supplied regarding the deposition of DNA sequences:

The ITS sequences are available at GenBank: MW911726, MW911727, and MW911729 to MW911735.

## Data Availability

The matrix of newly generated ITS sequences is available in the Supplemental Files.

The ITS matrix for phylogenetic analyses is available in the Supplemental Files and at TreeBASE: S28399. It is also available at Figshare: Xie, Meng-Le; Chen, Jun-Liang; Phukhamsakda, Chayanard; Dima, Bálint; Fu, Yong-Ping; Ji, Rui-Qing; et al. (2021): Cortinarius subsalor and C. tibeticisalor spp. nov., two new species from the section Delibuti from China. figshare. Dataset. https://doi.org/10.6084/m9.figshare.15982146.v1 https://treebase.org/treebase-web/search/study/summary.html?id=28399.

## New Species Registration

The following information was supplied regarding the registration of a newly described species:

Publication LSID: *Cortinarius subsalor* M.L. Xie, T.Z. Wei & Y. Li, sp. nov. MycoBank No. MB839320

*Cortinarius tibeticisalor* M.L. Xie, T.Z. Wei & Y. Li, sp. nov. MycoBank No. MB839321

## Supplemental Information

Supplemental information for this article can be found online at http://dx.doi.org/10.7717/peerj.11982#supplemental-information.

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
