# Peer review of "Cortinarius subsalor and C. tibeticisalor spp. nov., two new species from the section Delibuti from China"

_PeerJ, doi:10.7717/peerj.11982_

## Round 0.1 · original submission · Major Revisions

- I agree with both reviewers that the English needs editing. Please check carefully their corrections in the attached files, and if possible, find some advice from a fluent English speaker.

- Some sentences are a bit vague, such as that in lines 58-63: "the phylogenetic analyses verified and showed that there can still be some undescribed species in this section". (how can a phylogenetic analysis verify or show this? can you explain?). " In China, four
Delibuti species (...) were reported, but none of them have been confirmed by molecular sequence data" (again, explain how can these species be "confirmed" by molecular sequence data? You may rephrase saying that molecular sequence data confirms their distinctness, or separate position, or specific clade, for example). Please, check for clarity and consistency throughout the manuscript.


·

Basic reporting

In the present manuscript, Cortinarius subsalor and C. tibeticisalor belonging to the section Delibuti, are described from China as new to science, which is clear and unambiguous.However, 1) the English need to polish by a native speaker; 2) the latest literature references need to update; 3) in instruction part, the author need to add a paragraph according to the molecular systematics of this genus or section;

Experimental design

OK

Validity of the findings

1) Lack of the treebase datasets; 2) need add the MP analysis to this research;

Additional comments

The molecular data also support C.subsalor and C.tibeticisalor as new species. The phylogenetic analyses and biogeography of sect. Delibuti are discussed and a key to the species of this section currently known in the world is provided, which is good for supporting two new species. This research is significative to push the diversity of Chinese fungi. However, some commens also give to communicate:

1) there are many latest references missing about this genus in instruction part, the author have to add them;

2) please add the paragraph in instruction partabout the phylogeny of this genus;

3) add the BLAST result for top 10 taxa, which is easily get the identified accuracy, including the related 5 parameters;

4) this basidiospores data is ave, please add the holotype specimen dataset single in here, which is easily to get for reader.

5) this epithet need to change, in which both new species are similar to C. salor in same paper, which is more far-fetched;

6) add a distribution diagram of this group in China;

7) mark which is type specimen in this tree;

8) add the drawing picture for hyphae and basidiospores at least in Figs. 2 and 3;

other comments see the revised PDF.

·

Basic reporting

A few English sentences (or words) need to be modified. Please refer to the comments in the attachment.

Experimental design

No comment.

Validity of the findings

No comment.

Additional comments

The MS could be published after minor revisions.

The authors are suggested to to make the following revisions:
1)A few English sentences (or words) need to be modified. Please refer to the comments in the attachment.
2)Please make some checks and improvements to the "Key to species of sect. Delibuti". For example, the treatment of species "C. largodelibutus" seems not so good.
3)For other suggestions for improving the MS, please refer to the notes in the attachment.

---

## Round 0.2 · accepted · Accept

Dear Dr. Xie,

After reviewing your new version of your manuscript "Cortinarius subsalor and C. tibeticisalor spp. nov., two new species from the section Delibuti from China" I find it now improved and suitable for publication.

·

Basic reporting

In the present manuscript, two fungal species, Cortinarius subsalor and C. tibeticisalor belonging to the section Delibuti, are described from China as new to science. The comments on areas where the article have meet my standards with suggested improvements.

Experimental design

Good

Validity of the findings

ok

Additional comments

1) The drawing picture for hyphae and basidiospores in Figs. 2 and 3 are good.
2) It is better to add basidiospores data set of both holotype and AVE, but only holotype data.